# Electrospinning Silk Fibroin/Graphene Nanofiber Membrane Used for 3D Wearable Pressure Sensor

**DOI:** 10.3390/polym14183875

**Published:** 2022-09-16

**Authors:** Zulan Liu, Jiaxuan Wang, Qian Zhang, Zheng Li, Zhi Li, Lan Cheng, Fangyin Dai

**Affiliations:** 1State Key Laboratory of Silkworm Genome Biology, Key Laboratory of Sericultural Biology and Genetic Breeding, Ministry of Agriculture, Southwest University, Chongqing 400715, China; 2Chongqing Engineering Research Center of Biomaterial Fiber and Modern Textile, College of Sericulture, Textile and Biomass Sciences, Southwest University, Chongqing 400715, China

**Keywords:** silk fibroin, graphene, flexible sensor, electrospinning

## Abstract

With the improvement of science and technology, flexible sensors have become a hot research topic. Flexible sensors have broad application in human health detection and motion detection and other fields. In this paper, the silk fibroin/graphene nanofiber membranes were prepared by double needle electrospinning. In addition, the high sensitivity of the three-dimensional composite hierarchy was obtained by superimposing a monolayer silk fibroin/graphene nanofiber membrane, which was prepared via double needle electrospinning. In addition, the three-dimensional hierarchy was encapsulated by polydimethylsiloxane to prepare a pressure sensor. The sensitivity of the pressure sensor can achieve 7.7 Pa^−1^. In addition, this pressure sensor has excellent durability (>2000 cycles) and shorter response times (490 ms), which has broad research prospects in human health detection and motion detection.

## 1. Introduction

The sensor is very crucial as a man–machine interaction interface [1]. The physiological and motor data can be transmitted to computers in real time by sensors to analyze human health. With the development of the Internet of Things (IoT), conventional silicon-based devices with high Young’s modulus have become unsuitable for some applications increasingly [2], and wearable sensors are coming into view [3]. Internet of Things (IoT) refers to real-time acquisition of any object or process that needs monitoring, connection, and interaction through various devices and technologies such as information sensors, radio frequency identification technology, global positioning system, infrared sensors, laser scanners, and so on. Collect the information of sound, light, heat, electricity, mechanics, chemistry, biology, location and other needs, and realize the ubiquitous connection of things and things, things and people through all kinds of possible network access, so as to realize the intelligent perception, identification, and management of things and processes. In addition, with the improvement of science and technology, the health system has become personalized gradually. The Internet of Things can realize personalized and precise medical informatization, change the current medical environment, and provide more accurate and effective medical services [4,5,6]. The most advanced wearable sensors are light in weight and low in power consumption. They can be embedded not only in smart phones, but also in electronic watches and wristbands [7], such as the Apple Watch and Xiaomi wristband. These devices can monitor blood pressure, pulse, and other human health data [8]. However, due to the essential attributes of the rigid sensors, it can’t perfectly match the human body to obtain accurate physiological signals. Therefore, the flexible sensor has become the current research hot topic. The sensor with flexibility was very important in various applications such as electronic skins [9], wearable devices [10], and biomonitoring [11]. The human movement such as knee, elbow, finger, and so on can be monitored [12,13,14,15,16]. In addition, various other information of the human body, such as respiration, blood pressure, blood glucose, etc., can be detected. Therefore, in the personalized medicine field, the flexible pressure sensor also has broad development space [17,18] not just because of its detection performance but also for their simple structure, low production cost, and easy to read mechanism [19]. At present, the preparation methods of flexible pressure sensors can be divided into two categories: fill method and template method [20]. No matter what, the flexible pressure sensors always contain an active conductive material and a flexible substrate.

Graphene (Gr), a two-dimensional carbon material which consists of a single layer of SP [2] carbon atoms [21], with high conductivity, stable, physical, and chemical properties, is widely used in electrode materials, such as batteries, semiconductor devices [22], semiconductor [23], capacitors [24], and transistors [25]. Research shows that graphene also has a good piezoresistive effect and good conductivity, which makes graphene an excellent sensitive material in sensors [26]. Mamidi et al. [27] summarized the application including bioimaging, biosensing, wearable devices, and so on of carbonaceous nanomaterials and incorporated biomaterials. Lee et al. [28] prepared a flexible pressure sensor by using immersion to attach graphene to commercial fabrics. Lv et al. [29] prepared a stable and highly sensitive graphene/hydrogel strain sensor by using a two-step pouring method. The addition of glycerol not only enhanced the stability of the hydrogel, but also increased the stretchability of the hydrogel from 800% to 2000%. Li et al. [30] made two sensors by soaking cotton in graphene ink. The pressure sensors demonstrate excellent performance with high sensitivity (0.12–0.41 kPa ^−1^) and a wide operating range (0–20 kPa). The strain sensor showed excellent sensitivity (strain coefficients 22.6–83.7) and 27% large working strain.

Silk is a natural protein fiber, which possesses unique combined merits such as programmable biodegradability, good biocompatibility, and large-scale sustainable production. Compared with other biocompatible and biodegradable biomaterials, such as polycaprolactone (PCL), polylactic acid (PLA), poly(lactic-co-glycolic acid) (PLGA), etc., silk is a natural protein and is easy to obtain at a low cost. However, for natural biomaterials, silk has good regenerative properties and can be processed into various forms of materials [27]. Thus, silk has been popular for intensive design and study in flexible electronics over the past decade. In addition, with the development of fabrication techniques in material processing, silk can be engineered into a variety of forms according to the requirements, including silk fibers/textiles, nanofibers, films, hydrogels, and aerogels [31]. Wang et al. [32] reported a graphene oxide coated on silk fabric by vacuum filtration reduced by the hot press method, which provides the obtained silk fabric strain sensor with good piezoresistivity. In addition, the conductive silk fabrics are prepared by Liu et al. [33] via coating graphene oxide (GO) onto silk fabrics and followed by thermal reduction, which can be used for sensors, portable devices, and wearable electronic textiles. Ahmed et al. [34] summarized intrinsic properties and performance matrices of silk for integration with various nanomaterials and presented silk/nanomaterial-enabled bioelectronics applications, and in the end, future opportunities are also envisioned.

Electrospinning is a common technique for preparing nano-sized fibers. Electrospinning is possibly not as new as forcespinning (FS) or the centrifugal spinning method for the fabrication of nanofibers. However, the electrospinning technology is more mature and reliable. Furthermore, the prepared nanofibers are more uniform. In addition, the fiber orientation is adjustable [35]. Thus, in this paper, firstly, silk fibroin/graphene (SF/Gr) composite membranes were prepared by electrospinning. Then, a kind of three-dimensional (3D) structure SF/Gr flexible pressure sensor was obtained superimposing a monolayer SF/Gr nanofiber membrane and encapsulated by polydimethylsiloxane (PDMS) with good sensitivity and stability. In addition, SF/Gr sensor has great potential in human motion detection and human–computer interaction.

## 2. Experimental Section

### 2.1. Materials and Agents

Silk fibroin extracted from silkworm cocoons provided by the Southwest University. 1,1,1,3,3,3-Hexafluoro-2-propanol (HFIP, purity > 99.5%) was obtained from Aladdin Chemical Co. Ltd., Shanghai, China. Graphene (Gr, thickness: 1–3 layers, purity > 80%), was obtained from Aochuang technology Co. Ltd., Shenzhen, China. CH_3_CH_2_OH (Ethanol, purity > 99.5%) was obtained from Chuandong Chemical Co. Ltd., Chongqing, China. Polyvinylpyrrolidone (PVP, mean molecular weight: 1300 KDa) was obtained from Aladdin Chemical Co. Ltd., Shanghai, China.

### 2.2. Preparation

#### 2.2.1. Preparation of Silk Fibroin (SF)

The cocoons were cut into pieces and boiled twice in Na_2_CO_3_ (0.05 wt%) for 30 min each time. Then, they were washed with deionized water to remove the sericin before drying. Silk fibroin fibers were dissolved in CaCl_2_/EtOH/H_2_O solution (mole ratio, 1:2:8). After 5 days of dialysis, the SF solution was centrifuged at 8000 r/min for 10 min. Then, the desired SF solids were obtained by freeze-drying SF solution at −40 °C. The preparation process is shown in Figure 1a.

#### 2.2.2. Configuration of Electrospinning Solution

The weight of 3.2 g silk fibroin was dissolved in 20 mL of HFIP. Electrospinning solution A was obtained. The optimal dispersion ratio of graphene and PVP is 1:3 (As shown in Appendix A). Add different content (200 mg, 300 mg, 400 mg) graphene and (600 mg, 900 mg, 1200 mg) PVP to 20 mL of anhydrous ethanol. Then, sonication was applied for 2 h. Electrospinning solution B was obtained. The preparation process is shown in Figure 1a,b.

#### 2.2.3. Preparation of SF/Gr Nanofiber Membrane

Twenty milliliters of solution A (SF/HFIP) and 20 mL of solution B (Gr/PVP/ethanol) were put into two syringes for electrospinning at the same time. The distance between the syringe needle and the roller was 15 cm, and the syringe was propelled at a speed of 2.0 mL/h. Voltage parameters have positive voltage of 20 KV and negative voltage of 2 KV, respectively. The preparation process is shown in Figure 1c.

#### 2.2.4. Preparation of SF/Gr Sensor

The production process of SF/Gr sensor was divided into three steps. The first step was to extract silk fibroin from a silkworm cocoon. Then, electrospinning solution was prepared. Graphene was poured into anhydrous ethanol. The PVP was added as a dispersant and sonication for 2 h to make graphene disperse evenly in anhydrous ethanol. Secondly, SF/Gr nanofiber membrane was prepared by electrostatic spinning. Finally, SF/Gr nanofiber membrane was encapsulated by PDMS to obtain an SF/Gr flexible pressure sensor.

#### 2.2.5. Measurements and Characterization

The surface morphology of SF/Gr nanofiber membranes was analyzed by SEM (Phenom microscope, Holland, power at 5 KV). The porosity and diameter distribution of nanofibers were analyzed using the software ImageJ (National Institutes of Health, Bethesda, MD, USA) by SEM images. In addition, the water contact angle of nanofiber membranes was characterized by an Optical Contact Angle Tester (OCA15EC, Dataphysics, Filderstadt, Germany). The thermostability was performed on TG 209F3 Thermogravimetric Analyzer from NETZSCH (the results were shown in Appendix A). The composition of nanofiber membrane was characterized by Fourier transform infrared spectroscopy (FTIR, Spectrum 100, PerkinElmer, Waltham, MA, USA). The mechanical properties of the nanofiber membranes and sensor were tested with a universal stretcher at 25 °C and 60 ± 5% humidity. The electrochemical workstation was used to test the sensor performance under different deformations (CHI660E instrument, Shanghai, China).

## 3. Results and Discussion

### 3.1. Structural Characterization

The SEM images of SF/Gr nanofiber membranes were shown in Figure 1. The surface of SF nanofiber membrane was smooth, and the fibers were uniform. After electrospinning, the dispersion liquid of graphene/PVP in another needle, PVP, and graphene sheets were present on the SF fibers. When the concentrations of graphene and PVP were 10 mg/mL and 30 mg/mL, respectively, the bead structure appeared in fiber networks. When the concentration of graphene increased, the bead structure disappeared. With the increase of graphene concentration and PVP concentration, the greater number of graphene sheets attached to the three-dimensional network, and the denser the graphene sheets are. PVP and SF constructed a new three-dimensional network with graphene as the conductive material of the three-dimensional network.

As shown in Figure 2, the porosity of nanofiber membranes was increased after adding Gr. With the increasing of Gr concentration, the porosity was decreased gradually due to the Gr filling the pore between nanofibers. The nanofiber average diameter of SF, SF/Gr 10 mg/mL, SF/Gr 15 mg/mL, and SF/Gr 20 mg/mL were 2.17 ± 0.33, 1.13 ± 0.29, 1.59 ± 0.46, and 1.25 ± 0.19 μm. The nanofiber diameter of membranes was reduced after introducing Gr. However, the diameter of membranes with different Gr concentration showed little difference.

The water contact angle of nanofiber membranes was tested, and the results were shown in Figure 3. The water contact angle of SF, SF/Gr 10 mg/mL, SF/Gr 15 mg/mL, and SF/Gr 20 mg/mL were 0°, 14.9°, 19.8°, and 30.8°, which increased with the increase of Gr concentration. It suggested that the pure SF nanofiber membrane was hydrophilic, and with the introduction of hydrophobic Gr, the water contact angle was increased gradually.

Four membranes were tested by the infrared spectrometer (Figure 4). Silk fibroin has three most characteristic amide regions in the infrared spectrum, amide I (1700–1600 cm^−1^), amide II (1600–1500 cm^−1^), and amide III (1350–1200 cm^−1^), respectively. As can be seen from Figure 3, the peaks of amide Ⅰ, amide Ⅱ, and amide Ⅲ appeared at 1650 cm^−1^, 1517 cm^−1^, and 1240 cm^−1^. There are three characteristic peaks of PVP, which should appear in the vicinity of 1290 cm^−1^, 1660 cm^−1^ and 1463 cm^−1^, respectively. They are attributed to the stretching vibration of CN, C=O, and CH_2_ bond, respectively. As can be seen from Figure 4, SF/Gr 10 mg/mL, SF/Gr 15 mg/mL, and SF/Gr 20 mg/mL showed characteristic peaks at 1286 cm^−1^. The other two characteristic peaks may not appear because silk fibroin protein has the same groups in the same position. This may lead to the fusion of the peaks. It can be seen that the addition of graphene and PVP has almost no effect on the structure of silk fibroin. Graphene has almost no activity in the infrared spectrum. It can be judged that PVP and graphene are physically bound to silk fiber.

The mechanical properties of the four nanofiber membranes are shown in Figure 5. The ultimate stress and ultimate strain of SF nanofiber membrane are measured to be 1.38 MPa, 5.01% respectively. By adding the Gr component to the SF, the ultimate stress of nanofiber membranes was increased, while the ultimate strain of nanofiber membranes was decreased. This trend is common for composites which contain nanoscale reinforcing components with good dispersion and form strong interfacial interactions [36]. While the Gr concentration was 20 mg/mL, the ultimate stress was decreased compared to the SF/Gr 10 mg/mL and SF/Gr 15 mg/mL nanofiber membranes. It is indicated that, if the Gr concentration exceeds a certain amount, it will cause a weak knot between nanofibers. It is noted that the elongation of nanofiber membranes was decreased gradually with the increasing of Gr concentration.

### 3.2. Sensing Performance of SF/Gr Sensor

The sensing performance of pressure sensors can be evaluated by response time, sensitivity, stability, etc. The sensitivity test system consists of two parts. The first part is the pressure applying system, which is supplied by a universal stretcher machine. The second part is an electrical signal monitoring system, which consists of an electrochemistry workstation and computer. When the SF/Gr sensor is subjected to pressure, the transmitted electrical signal changes with the change of resistance due to the deformation of the three-dimensional network structure inside the sensor. The sensitivity (*S*) is calculated as *S* = (Δ*I/I_0_*)/Δ*P*. *I*_0_ is the initial current of the sensor under no pressure. Δ*I* is the relative change of current when the sensor is subjected to pressure. Δ*P* is applied pressure. As shown in Figure 6, the curve of current response and applied pressure was plotted point by point. In addition, the curve of current changing with pressure was obtained. The slope of the curve is the sensitivity of the sensor. SF/Gr was prepared with 10 mg/mL graphene, and the SF/Gr sensor consists of only one SF/Gr nanofiber membrane, which was named Gr10 mg-1. SF/Gr was prepared with 10 mg/mL graphene, and the SF/Gr sensor consists of two SF/Gr nanofiber membranes, which were called Gr10 mg-2. Other samples that were named follow this rule. These samples—Gr10 mg-1, Gr10 mg-2, Gr10 mg-3, Gr15 mg-1, Gr15 mg-2, Gr15 mg-3, Gr20 mg-1, Gr20 mg-2, and Gr20 mg-3 were tested. As can be seen from the figures, the relative change of current and pressure is linear, and the sensor prepared by stacking three layers of low-concentration graphene fiber membrane has the highest sensitivity, which can reach 7.7 Pa^−1^. Therefore, the SF/Gr sensor named Gr10 mg-3 was selected for follow-up research.

More tests were conducted on the SF/Gr sensor named Gr10 mg-3. The SF/Gr sensor not only has good sensitivity and a large detection range, but also demonstrated a fast response time. The sensor is attached to the wrist, and the response time of the sensor can be measured by the change of transmitted electrical signal by the sensor as the wrist moves. According to Figure 7a, the response time of the sensor was 490 ms, which suggested that the electrical signal of sensor can quickly change as the pressure changes. Most importantly, the SF/Gr sensor can maintain a short response time even after multiple deformations. To verify the reliability of the SF/Gr sensor, dynamic pressure is applied to the sensor. With the increase of pressure, the relative current output of the sensor also increased gradually (Figure 7b). To test the long-term stability of the SF/Gr sensor, the pressure was pressed 2000 times cyclically at 41.3 KPa (Figure 7c).

### 3.3. Human Motion Detection of SF/Gr Sensor

Human motion is an important signal of human health, which can be detected by wearable sensors via information collection and processing, and real-time health information of the body. It can evaluate whether the joint movement of human body is normal by connecting with the online doctor, in order to achieve the purpose of health care. The SF/Gr sensor prepared in this paper has advantages in human movement detection. It can be used for real-time movement detection of human by the contacting of irregular human skin, so as to provide the data for determining whether human joints are healthy. Figure 8a,b showed that the SF/Gr sensor has good bendability and stretchability, indicating that the sensor can adapt to various forms of deformation. Therefore, the sensor is applied to detect real-time information of the elbow, wrist, knuckle joint, and knee. As shown in Figure 8, the sensor can detect large bending motions in the elbow and knee (Figure 8d,g), as well as small bending motions in the finger joints and wrist (Figure 8c,e). Four different deformations presented different I-T curves, indicating that SF/Gr sensors can detect and distinguish different deformations sensitively.

### 3.4. Structure and Mechanism

The cross section optical image and SEM image of different layers SF/Gr sensor are shown in Figure 9. The encapsulated sensor has a PDMS-SF/Gr-PDMS sandwich structure. The upper and bottom layers are PDMS, which protect the internal network of the sensor and make the sensor have biocompatibility with human skin. The three-dimensional network that was constructed by PVP and SF can be observed from SEM images. As the carrier of the conductive material graphene, the three-dimensional network is deformed under pressure, resulting in changes in the contact resistance between the graphene and resulting in changes in sensor resistance. This structure is the working principle of the SF/Gr sensor. 

The SF/Gr pressure sensor model is shown in the figure below. The total resistance (*R*) of the sensor circuit consists of the volume resistance (*R_v_*) of the graphene sheet, the resistance of silk fiber and PVP fiber (*R_GP_*), and the contact resistance between the graphene sheets (*R_C_*). Copper foil is a conductive metal, and its resistance can be ignored. The calculation formula is as follows:*R* = *R*_*v*_ + *R*_*GP*_ + *R*_*C*_(1)

Because graphene sheets are protected by the PDMS packaging layer, the volume of graphene sheets will not change, so *R_v_* is constant. SF/Gr sensor is based on silk fiber and PVP fiber which have constant resistance. When the sensor is under pressure, the three-dimensional network, which is composed of silk fiber and PVP fiber, almost withstands all pressures. The three-dimensional network was deformed under the stress, which causes the graphene to contact each other and changes the contact resistance of graphene (*R_C_*). Therefore, it can be approximated that the change of the total resistance is caused by contact resistance of graphene. The sensor circuit can be modeled (Figure 10). The sensor is composed of one SF/Gr nanofiber membrane, as shown in Figure 10b. In the non-pressure case, the amount of graphene sheet that has been contacted is very small, so the sensor resistance is very high. When an external force is applied, the network of fibers is deformed, and the graphene sheets come into contact, resulting in a change in electrical resistance. The sensitivity is calculated by *S* = (*I* − *I*_0_/*I*_0_)/Δ*P*. With the increase of graphene concentration, the graphene sheets dispersed more densely in the fiber network and the more graphene was contacted under pressure. In addition, the resistance value was decreased. The *I* was increased, while *I*_0_ was close to 0. Therefore, the sensitivity showed an increasing trend. The sensitivity variation trend of Gr10 mg-1, Gr15 mg-1, and Gr20 mg-1 could prove this conclusion (Figure 6). When the sensor is composed of multiple SF/Gr nanofiber membranes, as shown in Figure 10c, the layers are in contact with the graphene sheets between the layers, resulting in an increase in *I*_0_. If the graphene sheets of the sensor are less distributed, its initial current I_0_ is considered to be close to 0 under no pressure, while Δ*I* changed a lot under pressure. This hypothesis can be proved by the sensitivity variation trend of Gr10 mg-2, Gr10 mg-3, Gr15 mg-2, and Gr20 mg-3 (Figure 6).

## 4. Conclusions

In this paper, firstly, silk fibroin/graphene (SF/Gr) composite membranes were prepared by electrospinning. The morphology, porosity, water contact angle, and diameter distribution of SF/Gr nanofiber membranes were characterized. The results suggested that SF/Gr nanofiber membranes possess good mechanical properties to make into 3D flexible sensors. Thus, the graphene/silk fibroin nanofiber membrane was packaged by PDMS into a pressure sensor. The prepared SF/Gr pressure sensor has a sensitivity of 7.7 Pa^−1^ and a response time as low as 490 ms. Benefitting from a stable three-dimensional network structure and a three-dimensional conductive network formed by well-dispersed graphene sheets and a PDMS package, the sensor maintained stable sensitivity under 2000 pressure cycles. The SF/Gr pressure sensor has great development potential in the field of wearable flexible sensors, which provides a new idea for the preparation of sensors.

## Data Availability

The data presented in this study are available on request from the corresponding authors.

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
