# Peer review of "Electrospinning Silk Fibroin/Graphene Nanofiber Membrane Used for 3D Wearable Pressure Sensor"

_polymers, 2022, doi:10.3390/polym14183875_

Round 1
Reviewer 1 Report
The aim of the study was to prepare by electrospinning a flexible pressure sensor with a 3D structure by applying a monolayer silk fibroin/graphene nanofiber membrane and encapsulated by polydimethylsiloxane with good sensitivity and stability. The manuscript is well written and contains only few errors (some highlighted below). The paper deals with an interesting topic, however there is no clearly presented novelty for the Readers. Could the Authors present a more substantial novelty of the work?
1. Abbreviations in the abstract should be minimized and only used when they are repeated in the abstract text. Please move unused abbreviations to the main text where they should be explained only once at the point of first use.
2. There should be an explanation in the text (Introduction) for the term "the Internet of Things (IoT)" as not everyone commonly uses the collective network of connected devices and the technology that facilitates communication between devices and the cloud, as well as between the devices themselves.
3. “mean molecular weight:1300000”; Please use variables and units in accordance with the SI system.
4. “In this paper, we prepared a kind of 3D structure degradable flexible pressure sensor”; Since the Authors do not study degradation and some parts of the sensor are not degradable, please do not use the word degradable in this sentence. Also, personal pronouns like "we" should not be used in the text. Statements should be impersonal. Please correct.
5. “Electrospinning Silk fibroin/Graphene Nanofiber Membrane used for 3D Wearable Pressure Sensor”; If the title is written in capital letters, only the conjunctions and prepositions appearing inside should be written in lowercase. “fibroin” and “used” should be capitalized. Please correct.
6. There should be a dot after the abbreviation "et al". Please correct.
7. "thickness: 1-3 layers"; There should be a space after the colon. Please correct throughout the text.
8. There should be a space before and after the mathematical operators (=, -, >, etc.). Please correct throughout the text.
9. According to SI variables should be in italics. Please correct throughout the text.
Reviewer 2 Report
The authors have prepared a research article entitled “Electrospinning Silk fibroin/Graphene Nanofiber Membrane used for 3D Wearable Pressure Sensor”. For this, Silk/GO composite fibers were primed by using double needle electrospinning. The obtained fibers were characterized and investigated their sensor capability. The Silk/GO/PDMS fibers exhibited improved sensitivity under the experimental conditions. The article has some interesting results and the authors have made considerable attention to preparing it. However, some issues need to be clarified before further consideration. Thus, the reviewer recommends this work can be published in Polymers after a major review.
1. The abstract is tediously long, and it should be modified. Remove very generic information about Silk and GO, and directly focus on the original finding and results. Particularly, modify the following text
“With the improvement of science and technology, flexible sensors have become a hot research topic. Flexible sensors have broad applications in human health detection and motion detection and other filed. Graphene (Gr) has excellent electrical properties and good piezoresistive properties, which makes it a good choice as sensitive material in sensors. Silk fibroin (SF) is a naturally protein derived from Bombyx mori silkworms, which is adequate and reproducible. It can be used as the nanofiber substrate via electrospinning”
2. There are several English grammar mistakes and typos, the language should be checked carefully throughout the manuscript
3. A schematic cartoon should be reported to understand the main content of the manuscript
4. Porosity, TGA, water contact angle, DSC, and diameter distribution of nanofibers are key characteristics of fiber systems. So, it is suggested to measure those properties.
5. Mechanical properties of nanofibers are key characteristics for biosensors/sensors, it is recommended to measure the tensile properties of fibers
6. The introduction is very short and should be improved entirely so that the reader can identify the scientific progress of this work.
7. There are several biocompatible and biodegradable biomaterials, including BSA, Gelatin, Zein, PCL, PLA, chitosan, UHMWPE, etc. but why did the authors select only Silk and GO? The authors should highlight the favorable characteristics that made Silk a more potent choice of materials for this study. Thus, the authors should emphasize why the Silk is familiar, or favorable compared to other biomaterials by using the following literature. Moreover, the information on biomaterials (BSA, Gelatin, Zein, PCL, PLA, chitosan, UHMWPE, etc.) should be explored in the introduction. Thus, the following articles should be quoted in the introduction and other sections.
https://doi.org/10.1016/j.compositesb.2022.110150
https://doi.org/10.3390/pharmaceutics14061116
It would be more realistic to cover such kind of research work in the current manuscript. Which will enrich the quality of the current manuscript as well as inquisitiveness to the readers.
8. Even though, there are several techniques are available to produce Nanofibrous scaffolds why did the authors select specifically electrospinning? For instance, the authors should compare their electrospinning method with Forcespinning® by quoting the following references: https://doi.org/10.3390/pharmaceutics14061116
9. According to the corrections, the conclusions may be modified.
Round 2
Reviewer 2 Report
The authors revised the manuscript. However, some of the latest works that focus on electrospinning are still missing to cover. For instance, the following review article can be quoted in the introduction of the revised manuscript.
https://doi.org/10.1021/acsbiomaterials.2c00786
The manuscript can be accepted after the addition of the above reference.